# Characterization of Dof Transcription Factors and the Heat-Tolerant Function of *PeDof-11* in Passion Fruit (*Passiflora edulis*)

**DOI:** 10.3390/ijms241512091

**Published:** 2023-07-28

**Authors:** Ge Chen, Yi Xu, Jie Gui, Yongcai Huang, Funing Ma, Wenhua Wu, Te Han, Wenwu Qiu, Liu Yang, Shun Song

**Affiliations:** 1Guangxi Academy of Agricultural Sciences/Guangxi Crop Genetic Improvement and Biotechnology Lab, Nanning 530007, China; bijiaya@163.com (G.C.); guijie142429@163.com (J.G.); huangyongcai1981@126.com (Y.H.); qww125@163.com (W.Q.); 2National Key Laboratory for Tropical Crop Breeding/Tropical Crops Genetic Resources Institute, CATAS/Germplasm Repository of Passiflora, Haikou 571101, China; lukydog163@163.com (Y.X.); mafuning@catas.cn (F.M.); w2410723376@163.com (W.W.); hante221@163.com (T.H.); 3College of Horticulture, Nanjing Agricultural University, Nanjing 210018, China; 4Hainan Key Laboratory for Biosafety Monitoring and Molecular Breeding in Off-Season Reproduction Regions, Sanya Research Institute, CATAS, Sanya 572000, China

**Keywords:** Dof transcription factor, passion fruit, high-temperature stress, gene expression, transgenic

## Abstract

Abiotic stress is the focus of passion fruit research since it harms the industry, in which high temperature is an important influencing factor. Dof transcription factors (TFs) act as essential regulators in stress conditions. TFs can protect against abiotic stress via a variety of biological processes. There is yet to be published a systematic study of the Dof (*PeDof*) family of passion fruit. This study discovered 13 *PeDof* family members by using high-quality genomes, and the members of this characterization were identified by bioinformatics. Transcriptome sequencing and qRT-PCR were used to analyze the induced expression of *PeDof*s under high-temperature stress during three periods, in which *PeDof-11* was significantly induced with high expression. *PeDof-11* was then chosen and converted into yeast, tobacco, and Arabidopsis, with the findings demonstrating that *PeDof-11* could significantly respond to high-temperature stress. This research lays the groundwork for a better understanding of *PeDof* gene regulation under high-temperature stress.

## 1. Introduction

There are about 520 species of passion fruit, which is a tropical and subtropical fruit [1]. The passion fruits grown in East Asia are mainly two varieties of *Passiflora edulis* (Sims and Degener), of which Sims is the most common and is the dominant variety, including yellow and purple skins [2]. Passion fruit is widely grown throughout tropical areas with less advanced economies and technological infrastructure, has a great therapeutic value, and is nutritious [3]. Abiotic stress is an important limiting factor for the healthy growth of passion fruit, especially heat stress, which negatively affects the flowering and fruiting process, leading to flower abscission and non-growth of fruit, seriously affecting yields, with ambient temperatures above 40 °C leading to over 80% yield reduction [4]. Therefore, it is particularly important to address the impact of high-temperature stress on passion fruit production, and mining resistance genes, especially transcription factors with powerful functions, and studying their mechanisms of resistance action and applying them to biological breeding is an important avenue.

With respect to the Dof gene, a powerful transcription factor, recent research has demonstrated its important role in enhancing plant abiotic stress functions (heat, salt, and drought). For example, overexpression of the tomato *SLDOF25/26* genes can improve the tolerance to drought and salt stress in Arabidopsis [5]. *TDDF1*, which encodes a tomato Dof protein, has been shown to increase plant salt and drought tolerance [6]. With the overexpression of tomato DOF genes *SlCDF1/3* in Arabidopsis, transgenic plants showed enhanced drought and salt resistance [5]. The expression of Dof Daily Fluctuations 1 (*TDDF1*) improves tolerance to drought and salt stress in tomatoes [6]. Furthermore, the Dof gene plays an important role in controlling the development and germination of seeds and the flowering process of plants [7,8,9]. For example, in grapes, several Dof genes play functional roles during flower, berry, and seed development, indicating their importance in grape growth and development [10]. *SlDOF10* plays an important role in plant reproductive development, especially during fruit setting [11].

The Dof gene family has a highly conserved Dof domain at the N-terminus, which is composed of the common core sequence (AT)/AAAG and is organized as a C2C2-type zinc-finger-like motif [12]. More Dof family genes have been researched since the first Dof gene (*ZmDof1*), and with the advent of genome sequencing, Dof genes have been widely identified in higher plants, such as Arabidopsis (36), rice (30) [13], sorghum (18) [14], and potato (35) [15].

In view of the biological functions of *Dof* genes, we analyzed the members of the *PeDof* family with a view to obtaining whether the family is amplified/contracted in passion fruit and preliminarily resolving its relevance to environmental adaptation. The members of the *PeDof*s and their characteristics were obtained by various biological techniques on the basis of a high-quality genome, transcriptome sequencing, and qRT-PCR analysis of *PeDof*s expression patterns under high-temperature stress and genetic transformation model organisms to verify the heat-tolerant biological functions of *PeDof-11*. This lays a good foundation for studying the regulatory mechanism of *PeDof*s and also provides important genes for passion fruit biobreeding.

## 2. Results

### 2.1. Identification of Passion Fruit Dof Members and Information Analysis

In total, 13 passion fruit *PeDof*s were identified, ranging in length from 738 bp (*PeDof-9*, 245 amino acids, molecular weight 26,871.18 Da) to 1503 bp (*PeDof-7*, 500 amino acids, molecular weight 54,769.37 Da), with all PeDof proteins containing Ser, Thr and Tyr phosphorylation sites, and *PeDof* genes located in the Nucleus (Table 1). The phylogenetic tree (Figure 1A) revealed that *PeDof*s were divided into three groups, with 3 and 10 members in groups 1 and 3, respectively. Most Dof members were closely homologously clustered in passion fruit with Arabidopsis and rice; e.g., *PeDof-13* with *AtDof1.7*, *PeDof-6* with *AtDof3.5*, *PeDof-11* with *AtDof4.2*, *PeDof-9* with *AtDof4.7*, respectively, showed the closest homology. Chromosomal localization analysis revealed that 13 *PeDof* genes were scattered on the chromosome (Figure 1B), with *PeDof-1/2/3/4* localized in Chr1. Chr3,4,8 contained two *PeDof*s, and Chr2,6,7 contained one *PeDof*s, respectively. Protein interaction analysis revealed that the proteins interacting with *PeDof*s were mainly *HSI2* (high-level expression of sugar-inducible gene 2S), *ABA1* (abscisic acid 1), and *COL5* (constans-like) and so on, and their functions were mostly involved in plant resistance and development, which provided important clues for further functional studies on the regulation of interactions (Figure 1C). Most instances of *PeDof*s 3D-structure modeling were structurally similar in that they contained the extension chain (red part) (Figure 1D).

### 2.2. Analysis of the Structures of PeDof Motifs and Promoters

The conserved motifs and UTR (untranslated region) were predicted (Figure 2A). *PeDof-2/4* had six motifs, *PeDof-12* had five motifs, *PeDof-15/11* had two motifs, while *PeDof-3/9/10/13* had only one motif, respectively. *PeDof* members contain CDS and UTR, except *PeDof-2/4/6/8/10/11/12/13*, which only had introns. *PeDof-5/9* only had 5′UTR, and *PeDof-1/7* only had 3′UTR. All 13 *PeDof*s promoter (−2000 bp DNA sequences regions) had TATA and CATA motif structures and contained a large number of cis-acting elements related to abiotic stress and resistance-related hormone responses, such as abscisic acid, salicylic acid, low temperature and light response, MeJA, auxin, gibberellin response elements, etc. (Figure 2B, Appendix A). These were the focus of our attention.

### 2.3. Collinearity Analysis of PeDofs

All genes were displayed on nine chromosomes circle, four collinearity pairs were found, and each *PeDof* was associated with 1 paralogous gene (Figure 3A). As obtained by interspecific covariance analysis (Figure 3B), 12,406, 3118, 25,224, and 14,638 genes in Arabidopsis, rice, poplar, and grape, respectively, were collinear with genes in passion fruit. Among 13 members, *PeDof-4* has homology with the genes (*At1G07640.3*, *At2G28810.1*, *At2G37590.1*, *At5G02460.1, OS05T0112200-01*, *VIT_06s0004g04520.t01*, *VIT_08s0007g00180.t01*, *VIT_13s0019g01410.t01*, *PNT56298*, *PNT17733*, *PNT30495*, *PNT22915*, *PNT19272*). In addition, *PeDof-2/4/12* has one homologous gene in Arabidopsis, poplar, and grape, respectively.

### 2.4. Expression of PeDof in Different Tissues of Passion Fruit and under High-Temperature Stress

We have analyzed the expression of some *PeDof*s in the leaves, roots, stems, and fruits of passion fruit (Figure 4). Among them, 6 of them were mainly expressed in the fruit (*PeDof-1/5/6/10/12/13*). *PeDof-3/4/7/8/9/11* were mainly expressed in the root. The results showed that each gene was expressed in different parts.

Under high-temperature stress, most *PeDof* members except *PeDof4/5/10/13* could be induced, especially *PeDof-11*, which is more than 6.7-fold differentially expressed under the stress at 2 h. The qRT-PCR result also verified this finding. Therefore, we also chose this high-expression gene as the research object to further study the relationship between this gene and improving heat tolerance (Figure 5).

### 2.5. Analysis of PeDof-11 in Different Tissues of Passion Fruit and under High-Temperature Stress

For the high-temperature stress experiment, the pYES2-*PeDof-11* and pYES2 empty vectors (control) were translated into INVSC1. The modified yeast grew better at 50 °C when exposed to high temperatures. This shows that *PeDof-11* may have a function in high-temperature stress (Figure 6).

We then transiently transformed tobacco with the *PeDof-11* promoter (Figure 7). The *PeDof-11*p-transformed tobacco and the control were both treated with a 42 °C high-temperature stress treatment. As can be seen from the figure, under the control of the *PeDof-11* promoter, the GUS staining is the deepest under the high-temperature treatment for 4 h, indicating that the *PeDof-11* is induced to express by the high-temperature stress. The findings demonstrate that the gene was substantially induced during the four-hour treatments.

To determine the role of *PeDof-11* in the expression of various tissues of Arabidopsis and in heat stress response, we transformed the *PeDof-11* promoter and the gene into Arabidopsis, respectively, and two lines were selected with the experiment, separately (Figure 8). The extent of GUS staining reflects the expression of the *PeDof* gene driven by the *PeDof-11* promoter. As shown in Figure 8A, the leaves of transgenic Arabidopsis had the deepest GUS staining. It shows that the gene is expressed mainly in the leaves. The wild-type and transgenic plants (7 days old) grown in 23 °C MS medium were treated at 42 °C for 36 h, and the GUS staining was greater and concentrated mostly in the leaves. We performed a GUS enzyme activity test and discovered that GUS activities are 3.9-fold higher than the control. The Arabidopsis plants (20 days old) grown in an incubator at 23 °C were treated at 45 °C for 8 h. The results showed that the transgenic plants were phenotypically more heat-tolerant than those of the wild type after high-temperature stress and that the transgenic Arabidopsis grew significantly better than the wild type, with higher survival than that of the wild type, after being restored to the normal conditions of 23 °C for 4 days of incubation. The results show that *PeDof-11* was induced by high-temperature stress.

## 3. Discussion

With more and more gene families of species being mined and analyzed, many *Dof* genes have been cloned. And some studies have shown that *Dof*s play a role in regulating gene expression in response to abiotic stress, such as drought, cold, and heat stress. However, the identity and function of passion fruit *Dof*s have remained unknown. In this research, 13 Dof family members were identified in the passion fruit genome and showed possible roles in *PeDof*s responses to abiotic stresses.

### 3.1. Identification and Characteristics of PeDof Genes

The Dof gene family is a plant-specific transcription factor family. Members of Dof genes have been discovered in different species since the first Dof gene was discovered in maize [16]. In this research, 13 *PeDof* genes have been identified. This number is similar to the number of Dof reported in spinach (22) and is lower than that of rice (30) [13], wheat (96) [11], and soybean (78) [6]. This indicates a massive shrinkage of the Dof family in passion fruit, and the 13 preserved members appear to be particularly valuable. The theoretical isoelectric points (pI) of Dof proteins ranged from 4.75 to 9.32; this result is similar to that in spinach [17].

In plants, Dof genes typically form polygenic families. The fundamental driving mechanism in the development of Dof genes is assumed to be repetitive events. In poplars, for example, 49% of *PTRD* genes were discovered to be localized in segmental and tandem repeat areas [18]. In apples, 57 and 18 MDD genes, respectively, are found in segmental and tandem repeat regions [19]. The cucumber genome contains two pairs of tandem repeats and six pairs of segmental repeats [20]. In this study, there are 28, 25, 19, and 2 connections between *PeDof* family members and poplar, Arabidopsis, grape, and rice *Dof* members, respectively.

Exon–intron structural arrangement can reveal evolutionary links within gene families [21]. In the current study, the number of introns in members of the *PeDof*s ranged from 0 to 2, with most members having no introns, a result similar to that of many other plant species, including pigeonpea, cucumber, poplar, and pear [11,18,20,22].

### 3.2. Cis-Elements of PeDof Genes

Cis-acting components, such as plant hormones and stress response, play a vital part in the life cycle of plants [17]. Cis-regulatory elements found in the promoters of co-expressed genes are thought to have a role in gene activity regulation. The majority of cis-elements in the *PeDof* gene family were associated with responses to light, stress response, and hormones (Appendix A). The majority of cis-elements found in the *SoDof* gene family were connected to light response [23]. Passion fruit growth is strongly light-demanding, and the cis-acting element of *PeDof*s contains a large number of light components, which also provides a basis for conducting studies on the mechanism of *PeDof* and light response.

### 3.3. Potential Role of PeDof genes in Different Tissues

The expression profiles of genes under specific conditions are closely related to biological functions [24]. Gene expression in different tissues may be associated with the underlying function of the gene. In general, genes located in fruits, meristems, and other tissues may be involved in the development of organs and tissues. In this study, most of the *PeDof* genes were distributed in fruits and roots. This predicts that the function of genes may be related to fruit development and stress resistance. In spinach, six *SoDof*s are expressed at higher levels in flowers than in other tissues, suggesting that they may be involved in reproductive processes such as flowering and fertility through floral organ responses [17].

### 3.4. The Response of Dof Genes in Abiotic Stresses

*Dof* genes are linked to abiotic stress tolerance in plants, according to a growing body of studies. For instance, in *Brassica napus*, the *Dof* gene family is able to respond to different abiotic stresses [25]. Analysis of the cis-acting elements of *CcDof*s showed a predominance of abiotic stress response elements, suggesting that these genes may be associated with abiotic stresses [22]. Overexpression of the *Dof* gene *TDDF1* in tomato increased blooming time, and transgenic plants were also more drought and salt tolerant [13]. *OsDof27* could improve the heat tolerance of transgenic rice [26]. Overexpression of *Dof*s (*SlCDF1/F3*) greatly improved Arabidopsis drought and salt tolerance while also activating other stress-responsive genes including *COR15*, *RD29A*, and *ERD10* [12]. One of the Arabidopsis Dof genes called CDF3 was highly induced by drought, extreme temperatures, and abscisic acid treatment [27].

## 4. Materials and Methods

### 4.1. Identification of Dofs and Information in Passion Fruit

The *PeDof* members were analyzed by bioinformatics software, including gene sequence, protein sequence, protein iso-electric point (pI) values and molecular weight (MW), prediction of subcellular localization, gene conserved region analysis, promoter structure, collinearity and protein interaction regulation and so on [3,4,28,29]. The relevant software and websites are listed in Appendix A.

### 4.2. Plant Materials, Transcriptome, and qRT-PCR Analysis

Healthy and virus-free passion fruit seedlings of the purple fruit varieties were chosen. They were grown in the soil under a growth chamber (Panasonic, MLR-352) (30 °C; 200 μmol·m^−2^·s^−1^ light intensity; 12-h light/12-h dark cycle; 70% relative humidity) to a height of about 40 cm and with 8–10 functional leaves, which were subjected to high-temperature stress treatments [28]. The seedlings were grown at 45 °C for high-temperature stress treatment, with other conditions unchanged. The plants were subjected to high-temperature stress treatment for transcriptome analysis [3]. The samples from the high-temperature stress treatment were utilized for RNA sequencing. The Biomic Biotechnology company (Beijing, China) was entrusted with sequencing services. In the additional information, the FRKP data of the transcriptome are shown in Appendix A. And we have frozen samples of roots, stems, fruits, and leaves for gene expression analysis in passion fruit tissues. The qRT-PCR analysis of *PeDof* genes was undertaken using suitable equipment (Light 96, Roche, Basel, Switzerland) [29]. Relative expression levels were calculated using the 2^−∆∆Ct^ method and normalized to the *PeDof*s. The date of qRT-PCR is shown in Appendix A.

### 4.3. Cloning and Vector Construction of PeDof-11 and the Promoter

The full-length cDNA and the promoter fragment of *PeDof-11* were amplified by PCR from the passion fruit (purple fruit varieties) and then cloned into the pCAMBIA1304 vector called pCAMBIA1304-*PeDof-11* and pCAMBIA1304-*PeDof-11*p, respectively. pCAMBIA1304-*PeDof-11*p was used for the tissue specificity and high-temperature stress experiments in the tobacco and Arabidopsis thaliana seedlings (7 days old). pCAMBIA1304-*PeDof-11* was used for the high-temperature stress experiments in the Arabidopsis thaliana (30 days old). The pYES2-*PeDof-11* vector was used for the yeast experiments.

### 4.4. Functional Complementation of PeDof-11 in Yeast (Saccharomyces cerevisiae)

The INVSc1 strain (*Saccharomyces cerevisiae*) was transfected with the pYES2-*PeDof-11* and the pYES2 vector (control). To perform the yeast complementation assays, the yeast liquid contained pYES2–*PeDof-11*, and the pYES2 empty vectors were first cultured in SD-Ura liquid medium at 30 °C to an OD600 value of about 1.0. The yeast liquid was diluted with sterile water by factors of 1, 10^−1^, 10^−2^, and 10^−3^, cultivated in SD-Ura liquid medium at 30 °C, and subjected to high-temperature stress (42 °C for 0 h, 12 h, and 24 h) to perform the yeast complementation tests. Three days later, the growth of the yeast was observed. and photos were taken.

### 4.5. Plant Transformation and High-Temperature Treatment

The Agrobacterium transformed with pCAMBIA1304-*PeDof-11*p was shaken at 28 °C to OD 600 = 0.8-1.0. Transient expression tests were conducted on two-month-old tobacco leaves. After high-temperature stress treatment (30 °C, 40 °C, 50 °C treated for 2 h), for additional staining tests, leaf discs with a diameter of 0.5 cm were cut off. The 7-day-old transgenic Arabidopsis with pCAMBIA1304-*PeDof-11*p were treated at 23 °C and 42 °C for 36 h. The T3 generation of 20-day-old transgenic Arabidopsis with pCAMBIA1304-*PeDof-11* was treated with the high-temperature treatment. The plants were planted in vermiculite and treated in an incubator under 45 °C under the light condition for 8 h, replaced at 23 °C for 4 d to recover cultivation (200 μmol·m^−2^·s^−1^ light intensity; 8-h light/16-h dark cycle; 70% relative humidity), observed the survival rate of seedlings of different. Two lines of pCAMBIA1304-*PeDof-11*p (L1-p, L2-p) and pCAMBIA1304-*PeDof-11* (L1, L2) transgenic Arabidopsis were taken for correlation studies, respectively.

### 4.6. Detection of GUS Activity

The different tissues of transgenic Arabidopsis with pCAMBIA1304-*PeDof-11*p, the 7-day-old seedlings, and the transgenic tobacco under normal and high-temperature stress were stained for GUS enzyme activity testing [30].

## 5. Conclusions

*Dof*s are widely reported to be related to the abiotic stress resistance of plants. We have identified 13 *Dof* members from the genome of the passion fruit. Analyses such as the evolutionary tree, structural domains, promoter cis-acting elements, inter-species, and intra-species collinearity were completed. In this research, the transcriptome results of *PeDof* gene members under high-temperature stress were verified by qRT-PCR, and the results showed that most of the members were induced to express by high-temperature stress. One of the *Dof* genes (*PeDof-11*) was highly induced by high-temperature stress. Further testing of this gene revealed that it might improve the capacity of transgenic tobacco, Arabidopsis, and yeast to withstand heat stress. The findings provide a solid platform for future research into the ability of passion fruit to withstand abiotic stressors.

## Figures and Tables

**Figure 1 ijms-24-12091-f001:**
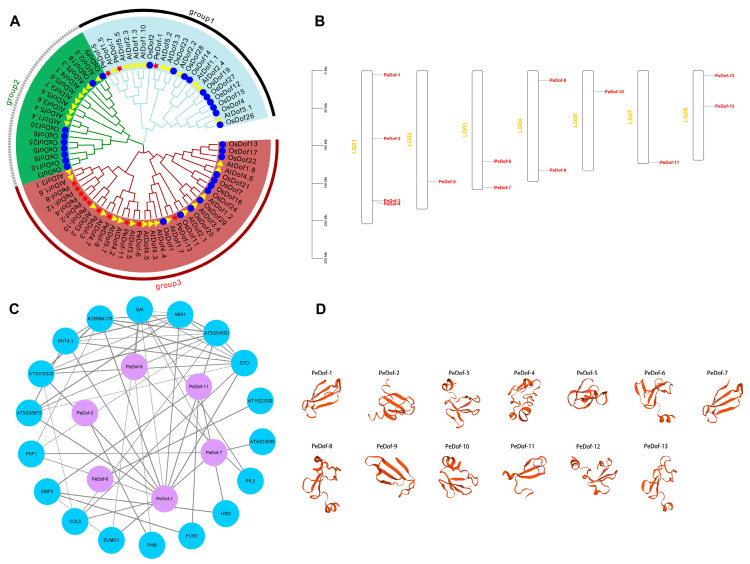
Homology analysis, chromosomal localization interaction regulated network, and protein structure prediction of PeDofs. (**A**) Phylogenetic relationship of *Dof*s among passion fruit, Arabidopsis and rice. red pentagram: passion fruit, yellow triangle: Arabidopsis, blue circle: rice (**B**) Distribution of 13 *PeDof* genes. (**C**) The predicted interaction networks of PeDof proteins. (**D**) The 3D-structure modeling of PeDof proteins.

**Figure 2 ijms-24-12091-f002:**
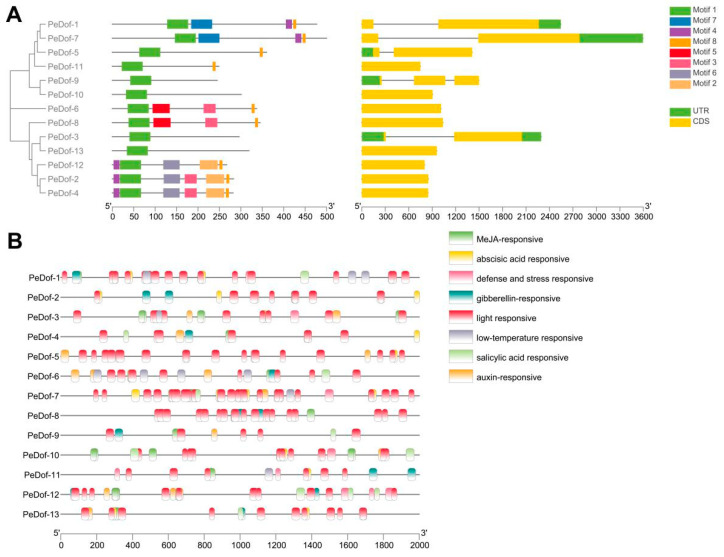
The *PeDof* genes and promoters structure characteristics analysis. (**A**) The clustering and gene structure analysis, (**B**) The promoter element analysis.

**Figure 3 ijms-24-12091-f003:**
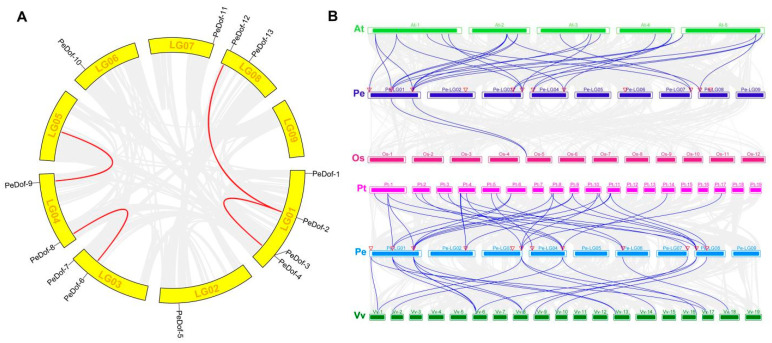
The collinearity distribution of *PeDof*s (**A**) and synteny collinearity of passion fruit, Arabidopsis, and rice genomes (**B**). The blue line represented the associated gene pairs.

**Figure 4 ijms-24-12091-f004:**
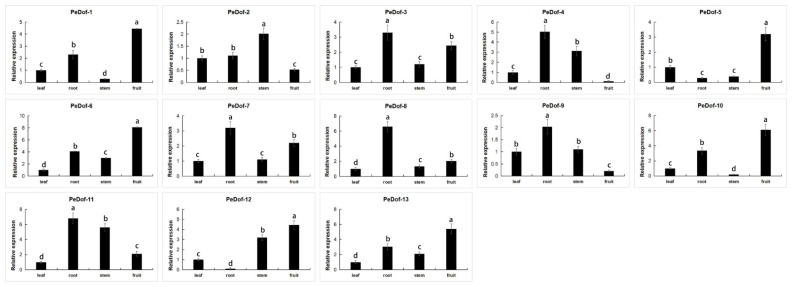
Expression analysis of the PeDofs during three fruit-ripening periods in the passion fruit. The details are shown in Appendix A. Data are means ± SD of *n* = 3 biological replicates. Means denoted by the same letter are not significantly different at *p* < 0.05 as determined by Duncan’s multiple range test.

**Figure 5 ijms-24-12091-f005:**
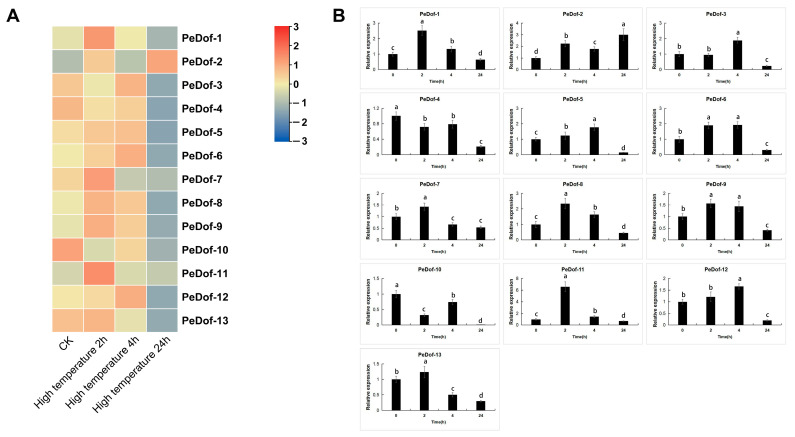
Transcriptome (**A**) and qRT-PCR CK: Plants without high-temperature stress treatment (**B**) of *PeDof*s responding to high temperature. Red and blue indicated high and low expression levels, respectively. Data are means ± SD of *n* = 3 biological replicates. Means denoted by the same letter are not significantly different at *p* < 0.05 as determined by Duncan’s multiple range test.

**Figure 6 ijms-24-12091-f006:**
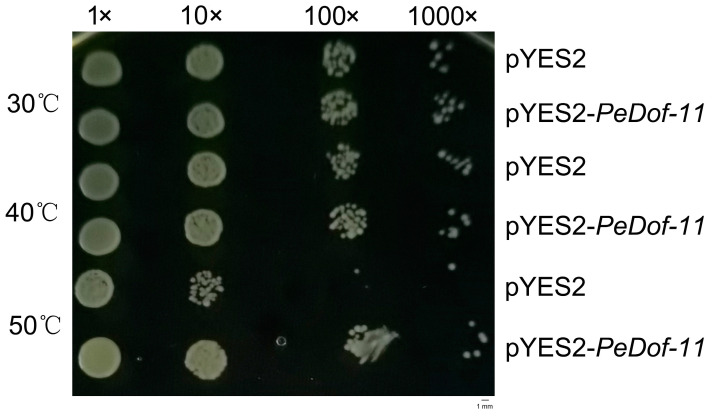
Growth status of the Saccharomyces cerevisiae INVSc1 Strain expressing pYES2–*PeDof-11* and pYES2 (control) under high-temperature stress.

**Figure 7 ijms-24-12091-f007:**
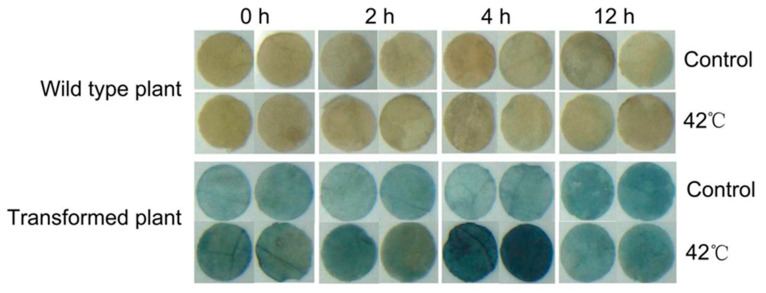
GUS staining was performed on transgenic tobacco. The tobacco was treated with 4 replicates per treatment and then processed into discs (diameter 0.5 cm).

**Figure 8 ijms-24-12091-f008:**
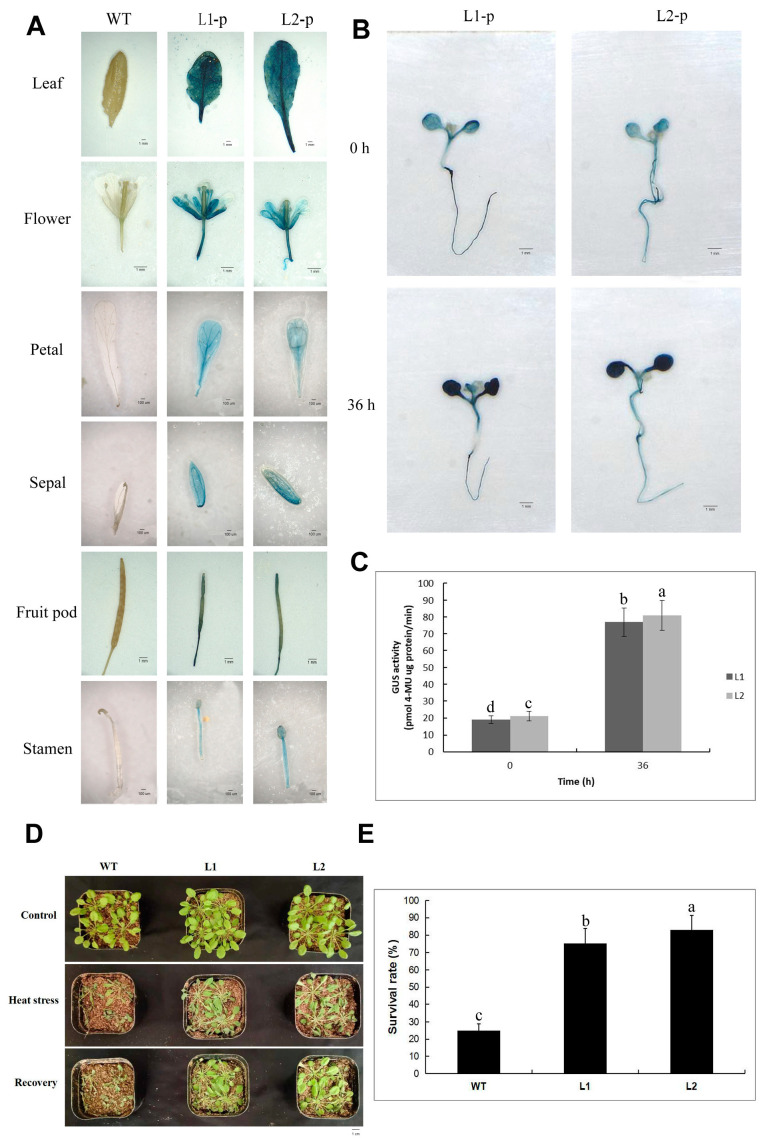
Tissue specificity and expression pattern of *PeDof-11* under high-temperature stress. (**A**) GUS staining in different tissues of transgenic Arabidopsis L1-p, L2-p. (**B**) GUS staining of transgenic Arabidopsis seedling. (**C**) GUS enzyme activity analysis of transgenic Arabidopsis (7 days old). (**D**) Grown seedling phenotype under high-temperature stress. (**E**) Grown seedling survival statistics of 20-days-old Arabidopsis. Data are means ± SD of *n* = 3 biological replicates. Means denoted by the same letter are not significantly different at *p* < 0.05 as determined by Duncan’s multiple range test.

**Table 1 ijms-24-12091-t001:** Analysis of physicochemical properties of *PeDof*s.

Gene	Gene ID	CDS Length (bp)	Protein Length (aa)	Molecular Formula	MW (Da)	pI	Number of Phosphate Sites	Subcellular Localization
*PeDof-1*	P_edulia010000233.g	1434	477	C_2245_H_3535_N_647_O_725_S_22_	51,894.98	6.72	Ser:56 Thr:22 Tyr:19	Nucleus
*PeDof-2*	P_edulia010001122.g	852	283	C_1289_H_1984_N_386_O_407_S_11_	29,752.93	9.25	Ser:39 Thr:13 Tyr:9	Nucleus
*PeDof-3*	P_edulia010002592.g	891	296	C_1396_H_2158_N_410_O_463_S_18_	32,670.04	6.31	Ser:35 Thr:21 Tyr:8	Nucleus
*PeDof-4*	P_edulia010002622.g	849	282	C_1274_H_1983_N_375_O_408_S_14_	29,529.87	8.96	Ser:42 Thr:13 Tyr:6	Nucleus
*PeDof-5*	P_edulia020006400.g	1083	360	C_1701_H_2678_N_490_O_553_S_19_	39,450.07	8.07	Ser:51 Thr:24 Tyr:13	Nucleus
*PeDof-6*	P_edulia030008409.g	1014	337	C_1564_H_2429_N_459_O_502_S_14_	36,143.10	9.31	Ser:49 Thr:17 Tyr:11	Nucleus
*PeDof-7*	P_edulia030009124.g	1503	500	C_2359_H_3647_N_679_O_782_S_23_	54,769.37	5.63	Ser:59 Thr:30 Tyr:22	Nucleus
*PeDof-8*	P_edulia040010004.g	1038	345	C_1599_H_2471_N_457_O_511_S_14_	36,721.80	8.85	Ser:51 Thr:18 Tyr:15	Nucleus
*PeDof-9*	P_edulia040010672.g	738	245	C_1179_H_1839_N_333_O_367_S_10_	26,871.18	9.01	Ser:27 Thr:19 Tyr:10	Nucleus
*PeDof-10*	P_edulia060015094.g	906	301	C_1437_H_2240_N_404_O_482_S_20_	33,529.21	4.75	Ser:44 Thr:14 Tyr:13	Nucleus
*PeDof-11*	P_edulia070018540.g	750	249	C_1138_H_1749_N_339_O_373_S_11_	26,500.11	8.14	Ser:32 Thr:12 Tyr:6	Nucleus
*PeDof-12*	P_edulia080018881.g	804	267	C_1237_H_1945_N_369_O_399_S_12_	28,755.00	9.28	Ser:40 Thr:15 Tyr:7	Nucleus
*PeDof-13*	P_edulia080019069.g	960	319	C_1501_H_2312_N_466_O_486_S_13_	35,078.48	8.57	Ser:44 Thr:14 Tyr:11	Nucleus

## Data Availability

The passion fruit genomic data and raw RNA-sequence data have been deposited in (https://ngdc.cncb.ac.cn/search/?dbId=gwh&q=GWHAZTM00000000), (https://ngdc.cncb.ac.cn/omix/release/OMIX563, accessed on 8 January 2020); the accession number is OMIX563-20-01, and the NCBI SRA number: SRP410034, and also accession number: SRR2240515-20.

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
