# Peer review of "Characterization of Dof Transcription Factors and the Heat-Tolerant Function of PeDof-11 in Passion Fruit (Passiflora edulis)"

_ijms, 2023, doi:10.3390/ijms241512091_

Round 1

Reviewer 1 Report

The authors tried to characterize PeDOFs related to their role in abiotic stress responses. Authors analyzed the sequences of PeDOF genes and their promoter regions, and also analyzed expression of those genes after incubation at high temperature. However, these analyses cannot provide meaningful information on the role of PeDOFs under high temperature conditions. Although PeDOF11 expression is clearly up-regulated after heat treatment, it is not enough to show whether PeDOF11 does indeed affect plant high temperature responses.

1. In Table 1, authors suggested that all PeDOFs would be located in the nucleus by analyzing protein sequence. However, it needs to be verified by experiments to clearly characterize PeDOFs. For example, analysis of GFP signals after transfection of vectors expressing PeDOFs-GFP fusion genes in passion fruit (or at least Arabidopsis) protoplasts is required to examine subcellular localization of these proteins.

2. In Results 2.4, the subtitle describes “expression of PeDOF in different tissues of Passion fruit”, but I cannot find gene expression data in different tissues. The displayed data seems expression of PeDOFs in a single tissue. For the analysis of tissue-dependent expression patterns, authors need to analyze PeDOF expression in leaves, stems, roots, flowers, and etc.

3. In Figure 4A, authors need to describe what “CK” means.

4. In Figure 6, it seems that authors used vectors expression GUS gene under the control of PeDOF11 promoter. However, there is no detailed description on the data.

5. In Figure 7, similar to the upper comment, authors should describe what GUS signal means in these experiments.

6. As the authors showed that PeDOF11 enhances heat tolerance in yeast, they need to examine whether PeDOF11 overexpression in Passion fruit or Arabidopsis improves resistance to heat stress to identify the role of PeDOF11 in plants.

Author Response

Thank you very much for your suggestions, we have carefully revised and added to the manuscript according to the comments.

  1. In Table 1, authors suggested that all PeDOFs would be located in the nucleus by analyzing protein sequence. However, it needs to be verified by experiments to clearly characterize PeDOFs. For example, analysis of GFP signals after transfection of vectors expressing PeDOFs-GFP fusion genes in passion fruit (or at least Arabidopsis) protoplasts is required to examine subcellular localization of these proteins.

Thank you very much for your suggestion, this part of our work is in progress, we will consider transient expression in plant protoplasts, or in tobacco for subcellular localisation experiments, and considering the completeness of the content of the subsequent manuscripts, we hope to include this part in the subsequent studies with in-depth functional analyses of Dof members.

  1. In Results 2.4, the subtitle describes “expression of PeDOF in different tissues of Passion fruit”, but I cannot find gene expression data in different tissues. The displayed data seems expression of PeDOFs in a single tissue. For the analysis of tissue-dependent expression patterns, authors need to analyze PeDOF expression in leaves, stems, roots, flowers, and etc.

Thank you very much for your suggestion, we have added the expression of PeDof family members in the roots, stems, fruits, and leaves of passion fruit.

  1. In Figure 4A, authors need to describe what “CK” means.

 “CK” means the plants without high temperature stress treatment, The corresponding content is supplemented in the legend of Figure 5. 

  1. In Figure 6, it seems that authors used vectors expression GUS gene under the control of PeDOF11 promoter. However, there is no detailed description on the data.

Thank you for your suggestion. The relevant description is added on lines 182 to 185.

  1. In Figure 7, similar to the upper comment, authors should describe what GUS signal means in these experiments.

Thank you for your suggestion. We have added the appropriate content in lines 192 to 193.

  1. As the authors showed that PeDOF11 enhances heat tolerance in yeast, they need to examine whether PeDOF11 overexpression in Passion fruit or Arabidopsis improves resistance to heat stress to identify the role of PeDOF11 in plants.

Thank you very much for your valuable suggestions. We have carried out relevant detection experiments of PeDof11 in transgenic Arabidopsis under high temperature stress in the previous period, but due to the consideration of the completeness of the content of the manuscript, we consider to add this content in the next in-depth study on the mechanism of action of PeDof11's heat resistance, including the detection of heat resistance in transgenic plants, physiological indexes and qRT-PCR assays, and screening and identification of reciprocal genes. In this manuscript, the phenotype as well as the survival rate of PeDof11 transgenic Arabidopsis under high temperature stress were added, and the experimental results showed that the gene could improve the heat tolerance of transgenic Arabidopsis.

Reviewer 2 Report

The article is devoted to the study of DOF transcription factors in the resistance of passion fruit to high temperature.

The Introduction should state more clearly why this particular object was chosen, as well as the need to study high temperature and the relationship with DOF TF.

The low quality of the presented figures does not allow a full assessment of their content, especially figures 3 and 4.

Methods need to be improved. How were the plants grown, in a greenhouse or climate chamber? If in a greenhouse, where she was, what season, altitude, humidity, sum of temperatures, length of day and night, etc. If climate chamber, what kind of lamps, brand, release date, spectral composition, color temperature, etc.

The transcriptome method is not described, the PCR method is not described. It is necessary to describe in detail the procedure for extracting nucleic acids from such a particular sample.

The technique of transformation of plants and yeasts is described very superficially.

How was it treated with high temperature, what was the humidity, was there wind or was the fan running, did the light and temperature act at the same time, or did the light turn off at the temperature?

Statistical methods are not described at all.

Conclusions are superficial

Author Response

Thank you very much for reviewing the manuscript and for your valuable comments, which we have changed and added accordingly. Most of the following issues are in the description of the methodology, and we have added the details accordingly.

1.The Introduction should state more clearly why this particular object was chosen, as well as the need to study high temperature and the relationship with DOF TF.

Thank you very much for your suggestions, we have made additions to the introductory section as well as adjustments to the order, please check.

2.The low quality of the presented figures does not allow a full assessment of their content, especially figures 3 and 4.

We have replaced all the figure insertion formats and uploaded the original figures, please check.

3.Methods need to be improved. How were the plants grown, in a greenhouse or climate chamber? If in a greenhouse, where she was, what season, altitude, humidity, sum of temperatures, length of day and night, etc. If climate chamber, what kind of lamps, brand, release date, spectral composition, color temperature, etc.

Thank you very much for your suggestion. Because the reference is cited, the specific parameters are not listed and have been added in lines 303 to 307, please check.

4.The transcriptome method is not described, the PCR method is not described. It is necessary to describe in detail the procedure for extracting nucleic acids from such a particular sample.

Thank you very much for your suggestion. Because the reference is cited, the specific parameters are not listed and have been added in part 4.2.

5.The technique of transformation of plants and yeasts is described very superficially.

Thank you for your suggestion. The transformation methods for plants and yeast have been added in part 4.4 and 4.5.

6.How was it treated with high temperature, what was the humidity, was there wind or was the fan running, did the light and temperature act at the same time, or did the light turn off at the temperature?

We refer to the reference(Qin et al.,2022), where plants were treated in an incubator at high temperatures and in light with 200 μmol·m−2·s−1 light intensity and 70% relative humidity.

Reference:

Qin QQ, Zhao YY, Zhang JJ, Chen L, Si WN and Jiang HY. A maize heat shock factor ZmHsf11 negatively regulates heat stress tolerance in transgenic plants. BMC Plant Biology 2022, 22:406.

7.Statistical methods are not described at all.

Thank you very much for pointing out the shortcomings, we have added the statistical methodology in the legends.

8.Conclusions are superficial

Thank you for your suggestion. The conclusions section has been added, please check.

Round 2

Reviewer 1 Report

The authors performed additional experiments to improve the quality of the manuscript. Although still subcellular localization analysis of PeDOFs in passion fruit is missing, they showed that introduction of PeDOF-11 enhances heat tolerance in Arabidopsis. However, detailed descriptions on these experiments are still not enough. I think the authors overexpressed PeDOF-11 in Arabidopsis, but I cannot find related descriptions in the main text (What is L1 and L2 mean?).

In addition, the revised manuscript is extremely confusing to me. For example, revised figures are overlapped on the previous versions. These unorganized formats should be corrected.

Author Response

Thank you for your review. We have made additions to the content based on the comments, as follows,

1.The authors performed additional experiments to improve the quality of the manuscript. Although still subcellular localization analysis of PeDOFs in passion fruit is missing, they showed that introduction of PeDOF-11 enhances heat tolerance in Arabidopsis. However, detailed descriptions on these experiments are still not enough. I think the authors overexpressed PeDOF-11 in Arabidopsis, but I cannot find related descriptions in the main text (What is L1 and L2 mean?).

Due to an oversight, we did not describe L1 and L2 clearly. L1 and L2 are two lines in pCAMBIA1304-PeDof-11p and pCAMBIA1304-PeDof-11 transgenic Arabidopsis, respectively. To distinguish between promoter and gene constructs of transgenic plants, we refer to pCAMBIA1304-PeDof-11p transgenic Arabidopsis strains as L1-p and L2-p, and pCAMBIA1304-PeDof-11 transgenic Arabidopsis strains as L1 and L2. The relevant changes are in line 229,351-353 and fig. 8.

2. In addition, the revised manuscript is extremely confusing to me. For example, revised figures are overlapped on the previous versions. These unorganized formats should be corrected.

Thanks for checking it out. We have checked the format of all the images and re-uploaded the originals, please check.